# Design and Immunoinformatic Assessment of Candidate Multivariant mRNA Vaccine Construct against Immune Escape Variants of SARS-CoV-2

**DOI:** 10.3390/polym14163263

**Published:** 2022-08-10

**Authors:** Mushtaq Hussain, Anusha Amanullah, Ayesha Aslam, Fozia Raza, Shabana Arzoo, Iffat Waqar Qureshi, Humera Waheed, Nusrat Jabeen, Sanya Shabbir, Muneeba Ahsan Sayeed, Saeed Quraishy

**Affiliations:** 1Bioinformatics and Molecular Medicine Research Group, Dow Research Institute of Biotechnology and Biomedical Sciences, Dow College of Biotechnology, Dow University of Health Sciences, Karachi 75330, Pakistan; 2Department of Microbiology, University of Karachi, Karachi 75270, Pakistan; 3Sindh Infectious Disease Hospital and Research Centre, Dow University of Health Sciences, Karachi 75330, Pakistan; 4Dow University of Health Sciences, Karachi 75330, Pakistan

**Keywords:** SARS-CoV-2, COVID-19, mRNA vaccine, immune evasion, immunoinformatics

## Abstract

To effectively counter the evolving threat of SARS-CoV-2 variants, modifications and/or redesigning of mRNA vaccine construct are essentially required. Herein, the design and immunoinformatic assessment of a candidate novel mRNA vaccine construct, DOW-21, are discussed. Briefly, immunologically important domains, N-terminal domain (NTD) and receptor binding domain (RBD), of the spike protein of SARS-CoV-2 variants of concern (VOCs) and variants of interest (VOIs) were assessed for sequence, structure, and epitope variations. Based on the assessment, a novel hypothetical NTD (h-NTD) and RBD (h-RBD) were designed to hold all overlapping immune escape variations. The construct sequence was then developed, where h-NTD and h-RBD were intervened by 10-mer gly-ala repeat and the terminals were flanked by regulatory sequences for better intracellular transportation and expression of the coding regions. The protein encoded by the construct holds structural attributes (RMSD NTD: 0.42 Å; RMSD RBD: 0.15 Å) found in the respective domains of SARS-CoV-2 immune escape variants. In addition, it provides coverage to the immunogenic sites of the respective domains found in SARS-CoV-2 variants. Later, the nucleotide sequence of the construct was optimized for GC ratio (56%) and microRNA binding sites to ensure smooth translation. Post-injection antibody titer was also predicted (~12000 AU) to be robust. In summary, the construct proposed in this study could potentially provide broad spectrum coverage in relation to SARS-CoV-2 immune escape variants.

## 1. Introduction

Mass vaccination against SARS-CoV-2 on a global scale noticeably reduces the disease burden and intensity of the pandemic, COVID-19, around the world. Several vaccines against the SARS-CoV-2 have successfully passed through clinical trials and were made available for mass vaccination [1,2,3,4,5]. These vaccines are primarily based on the inactivated virus, recombinant adenovirus, and viral spike protein mRNA [6]. The idea of the mRNA vaccines is around 20 years old and lately, empirical evidence for its efficacy and suitability has been demonstrated against several pathogens such as HIV-1, Zika virus, rabies, and influenza virus in animal models and humans [7,8,9,10]. So far, against SARS-CoV-2, two mRNA-based vaccines, BNT162b2 and mRNA-1273, have been developed by Pfizer and Moderna, respectively [1,3]. However, reduction in the neutralization of SARS-CoV-2 variants by the sera taken from mRNA-vaccinated individuals has been reported extensively. This raises concerns regarding the efficacy of the currently available repertoire of vaccines including mRNA vaccines against the emerging and circulating variants of SARS-CoV-2 [11,12,13,14,15,16,17,18,19,20,21,22,23,24].

During the COVID-19 pandemic, random mutations in the genome of SARS-CoV-2 led to the emergence of a number of genetic variants of the virus [25]. These genetic variations have been sculpted by the forces of natural selection resulting in the emergence of viral variants with improved transmissibility, higher virulence, and/or immune escape properties. These variants were categorized as variants of concern (VOC) and variants of interest (VOI) [26]. The first of so-called VOC was recognized in September 2020 in the UK (B.1.1.7) and dubbed as Alpha variant [27]. The variant showed a 50% high transmission rate, increased virulence, and reduced neutralization from post-vaccination sera compared to the SARS-CoV-2/Wuhan strain [12,28]. Later on, three other VOCs, B.1.351 (β), P.1 (γ), and B.1.617.2/AY.1-3 (∆) were also recognized with reduced neutralization of variants by post-vaccination sera [13,16,22,29,30]. Recently, on 26 November 2021, WHO declared another variant of SARS-CoV-2 of B.1.1.529 lineage as VOC, dubbed Omicron (ο), with a considerably greater number of mutations in the spike protein compared to all previously recognized VOCs [27]. Recently, infection from a new variant of the virus dubbed XE, of BA.1/BA.2 lineage has been reported in different parts of the world [26]. Owing to the extensive variations in the spike protein of SARS-CoV-2-ο, it has been shown to evade the acquired immunity gained by the previous infection or vaccination [24,31].

Antibodies isolated from the SARS-CoV-2 infected individuals bind with the N-terminal domain (NTD) and receptor binding domain (RBD) of the viral spike protein. Co-crystal complex structures of NTD and RBD with their respective antibodies have also been resolved [32,33]. The mRNA vaccine constructs solely based on the SARS-CoV-2 spike RBD domain have shown promising results in the murine and non-human primate models [34,35,36]. However, despite the attachment of the SARS-CoV-2 to the host cell is mediated by the RBD domain, it is becoming increasingly evident that the NTD region of the spike protein is critical for developing a robust immune response against the virus [37]. Therefore, it is not surprising that immune escape variants of SARS-CoV-2 harbor mutations both in the NTD and RBD of the spike protein [38]. Moreover, antibodies isolated from the plasma of individuals infected with one type of SARS-CoV-2 variant have shown cross-immune protection against other variants of the virus [39,40]. This suggests that new viral variants emerged due to the accumulation of the mutations that are exclusively present in the antecedent variants and can elicit a cross-protective immune response against multiple variants. Taken together, it is possible to design specific NTD and RBD to develop a novel mRNA construct that could concomitantly provide protection against multiple immune escape variants of SARS-CoV-2. Based on this notion, in this study, we present the design and immunoinformatic analyses of a novel mRNA construct as a candidate for mRNA vaccine against established immune escape variants of SARS-CoV-2 (VOCs and VOIs). This approach is novel in comparison to previously suggested methodologies based on the development of multi-spike protein vaccine construct or assembling modified RBDs of SARS-CoV-2 variants or altogether using multiple inactivated SARS-CoV-2 variants [41,42]. The approach led to the design of a novel mRNA construct of shorter size with broad coverage against immune escape variants of SARS-CoV-2.

## 2. Materials and Methods

Methodology of the present study is summarized schematically in Figure 1.

### 2.1. Sequence Retrieval

Whole genome sequences of SARS-CoV-2/Wuhan (NC_045512) were retrieved from National Center for Biotechnology Information (NCBI) genome database (https://www.ncbi.nlm.nih.gov/, accessed on 1 July 2022). A total of 10 whole genome sequences of each variant were retrieved from GISAID database (https://www.gisaid.org/, accessed on 1 July 2022) [25] to assess intra-variant variations in the spike protein sequence (See Appendix A). Genome sequences of known SARS-CoV-2 VOCs, α (B.1.1.7; EPI_ISL_728343), β (B.1.351; EPI_ISL_1381260), γ (P.1; EPI_ISL_1171658), ∆ (B.1.617.2; EPI_ISL_3446728), ο (B.1.1.529; EPI_ISL_6640919), οXE (BA.1/BA.2; EPI_ISL_12451987) and SARS-CoV-2 VOIs, μ (B.1.621; EPI_ISL_4036743) and λ (C.37; EPI_ISL_3460005) were taken as representative. Whole genome sequence of Epstein–Barr virus (NC_007605.1) was retrieved from NCBI genome database to later extract gly-ala linker sequence. For 5′-UTR, de novo designed sequence was used, whereas 3′-UTR region sequence was retrieved from murine Rps27a 3′-UTR augmented with sindbis virus RNA binding protein region, R3U [43]. Signal peptide RNA sequence was derived from human Igκ [44].

### 2.2. Multiple Sequence Alignment

To extract the NTDs and RBDs from SARS-CoV-2 variants spike proteins, full genome sequences of the variants were aligned with the genomic sequence corresponding to the whole length spike protein of SARS-CoV-2/Wuhan using CLUSTALW under default parameters [45]. Additionally, separate alignments were then developed for genome sequences corresponding to the NTDs and RBDs of the spike proteins of SARS-CoV-2 variants (See Appendix A). For linker, the gene sequence encoding EBNA1 protein was retrieved from the EBV genome using CDS function and later its gly-ala repeat region (112–121) was extracted by manual counting of the codons.

### 2.3. Design of Novel NTD and RBD

A novel hypothetical NTD (h-NTD) and RBD (h-RBD) domains were designed by assembling all potentially immune escape mutations (Figure 2) in the corresponding domains using Wuhan spike protein NTD and RBD as the backbone. In case of over lapping mutations, preferences were given to the variations found in VOCs over VOIs.

### 2.4. In Silico Translation and Sequence Variations

In silico translations of separated NTD and RBD sequences of SARS-CoV-2 variants and h-NTD and h-RBD were carried out using ExPasy translate tool (https://web.expasy.org/translate/, accessed on 1 July 2022) [46]. The protein sequences were then aligned with the NTD (13-304) and RBD (319-541) of SARS-CoV-2/Wuhan spike protein using CLUSTALW to ensure the presence of reported corresponding mutations. All alignments were visualized in CLC sequence viewer.

### 2.5. Molecular Modeling

Atomic coordinates of NTD (PDB: 7C2L) and RBD (PDB: 7K8S) of SARS-CoV-2/Wuhan were retrieved from RCSB database (https://www.rcsb.org/, accessed on 23 March 2022) [47]. The structure of gly-ala repeat was retrieved from the full-length structural model of EBNA1 [48]. The coordinates were then used to develop structural models of NTDs and RBDs of the spike protein of all SARS-CoV-2 immune escape variants, and h-NTD and h-RBD using Swiss-Model (https://swissmodel.expasy.org/, accessed on 1 July 2022) [49].

### 2.6. Structural Analysis

To explore the structural variations between the NTDs and RBDs, individual models of the NTDs and RBDs of SARS-CoV-2 immune escape variants, and h-NTD and h-RBD were superimposed over the resolved structures (PDB: 7C2L and 7K8S) using Swiss-PdbViewer v 4.1.0 [47,50]. The structural variations were estimated in terms of deviation in Cα backbone of the proteins in Å. Ramachandran plots of the models were developed using MolProbity (http://molprobity.biochem.duke.edu/, accessed on 1 July 2022) to monitor the distribution of dihedral angles [51]. Gibbs Free energy of all the modeled NTDs and RBDs were estimated using Swiss-PdbViewer v 4.1.0 in KJ/mol.

### 2.7. Epitope Mapping

Epitopes of the NTDs and RBDs of all SARS-CoV-2 immune escape variants, and h-NTD and h-RBD were predicted based on their three-dimensional structure using DiscoTope v2.018 (http://tools.iedb.org/discotope/, accessed on 1 July 2022) under default threshold score (−3.7) [52]. DiscoTope coalesces and accounts for the proportion of the amino acids, surface accessibility, and spatial orientation of the residues in structural conformation for the epitope prediction. For SARS-CoV-2/Wuhan originally resolved structures of NTD (PDB: 7C2L) and RBD (PDB: 7K8S) were used [47].

### 2.8. Molecular Modeling of Novel mRNA Construct

h-NTD and h-RBD protein sequences were first joined by 10 mer gly-ala repeat as encoded protein of the construct (DOW-21). Full-length molecular model of the molecule was then developed. The model was assessed for structural, thermodynamic, and epitope distribution as described in Section 2.6 and Section 2.7. All protein models, superimpositions, and epitopes were visualized in DS visualizer 2021.

### 2.9. Immune Response Simulation

Time course immune response for antibodies and cytokines against the protein encoded by candidate mRNA construct (DOW-21) was predicted by C-ImmSim server (https://kraken.iac.rm.cnr.it/C-IMMSIM/, accessed on 1 July 2022) [53]. For prediction, one injection comprised of 1000 mRNA constructs per unit was used.

### 2.10. mRNA Construct Sequence Assembly and Validation

Nucleotide sequences of SARS-CoV-2/Wuhan NTD and RBD were modified to correspond to protein sequences of h-NTD and h-RBD. The sequences were then verified by in silico translation using Expasy Translate Tool [46]. Full-length mRNA construct (DOW-21) was then developed by joining nucleotide sequences of the h-NTD and h-RBD intervened by sequence of gly-ala repeats (encoding 10 amino acids) derived from the EBNA1 protein of EBV. At the 5′ end, signal peptide sequence of human Igκ was added, and finally 5′-UTR and 3′-UTR sequences were incorporated at both 5′ and 3′ ends of the construct, respectively [43,44]. At 3′ end of the construct, a polyA tail of 120 nucleotides was manually added. In silico translation was carried out to verify that the designed construct is in frame. To improve the proportion of the GC content, preferably third base of the codons in the constructs was modified without invoking any substitution of amino acid.

### 2.11. MicroRNA Binding Sites Prediction

Expression of the mRNA constructs for vaccines could be modulated by its interaction with different human microRNA (miRNA) [54]. Therefore, full-length sequence of DOW-21 was screened for human miRNA binding sites using miRDB (http://mirdb.org/, accessed on 1 July 2022) [55]. The program searches the target mRNA sequence for the binding of 568 human precursors and 654 human mature miRNAs. The binding sites were predicted based on sequence homology and scored accordingly. All positive hits with target hit scores ≥ 90 were considered for the optimization of the construct. The respective codons holding miRNA binding sites were replaced with the alternate codons encoding same amino acids. The process was continued till all the miRNA binding sites (above the target score ≥ 90) were lost. Finally, the construct sequence was again verified by in silico translation.

### 2.12. Structure Analysis of mRNA Constructs

Structure and thermodynamic attributes of DOW-21 were predicted using RNA fold (http://rna.tbi.univie.ac.at/cgi-bin/RNAWebSuite/RNAfold.cgi/, accessed on 1 July 2022) under default parameters to calculate folding under Minimum Free Energy (MFE) with Partition Function (PF), avoiding isolated base pairs [56]. The bases in the nucleotide sequence of the construct were modified till lines in the mountain plot of MFE, PF, and centroid structure coalesced. The final sequence of the construct was again subjected to the in silico translation and checked for the miRNA binding sites. 

## 3. Results

### 3.1. Sequence Variations in NTDs and RBDs

Immunologically, NTD and RBD are the most important regions of the SARS-CoV-2 spike protein, as antibodies developed due to natural infection and/or vaccination are mainly targeting these domains of the protein [32,33,37]. Multiple sequence alignments of both NTDs and RBDs of immune escape variants of SARS-CoV-2 demonstrated considerable sequence conservation especially within the sequences of each variant (See Appendix A). Nevertheless, deletions were observed in NTDs of variants such as SARS-CoV-2-α (ΔH69-V70; ΔY144), and SARS-CoV-2-β (ΔL242-A244) compared to SARS-CoV-2/Wuhan. Compared to SARS-CoV-2/Wuhan, the newly recognized omicron variant of the SARS-CoV-2 also harbors several deletions (ΔH69-V70; ΔV143-Y145; ΔL211) and an insertion (Ins 214 EPE) in its NTD domain of the spike protein, in addition to five different amino acid substitutions (Figure 2a). In comparison to NTD, RBDs sequences were found relatively more conserved amongst SARS-CoV-2 variants, owing to its involvement in the binding with the host cell receptor [57]. However, few substitutions such as L452R, E484K, and N501Y are considered important in relation to the immune evasion of the SARS-CoV-2 variants [12,13,14,15,16,29,58]. SARS-CoV-2-α/β/γ/ο, all found to have N501Y substitution in their RBDs, whereas E484K was found in VOCs such as SARS-CoV-2-β/γ and VOI SARS-CoV-2-μ. However, in the SARS-CoV-2-∆ and SARS-CoV-2-ο, Glu-484 was found substituted by Gln or Ala, respectively. Another important variation, L452R/Q, was confined in SARS-CoV-2-∆ and SARS-CoV-2-λ (Figure 2b).

### 3.2. Structural Variations in NTDs and RBDs

Superimposition of NTDs of different SARS-CoV-2 variants over SARS-CoV-2/Wuhan showed 0.13–0.45 Å deviations in the Cα backbone suggesting overall structural conservation in the domain between different viral variants. However, subtle variations were observed in the turn present between β3 and β4 in SARS-CoV-2-α and SARS-CoV-2-ο, potentially due to the deletions (ΔH69-V70) found in the variants compared to SARS-CoV-2/Wuhan (Figure 3a). Like sequences, structures of RBD were also found relatively more conserved than NTD with Cα backbone deviation ranging from 0.12 Å to 0.14 Å between different SARS-CoV-2 variants compared to SARS-CoV-2/Wuhan (Figure 3b). In comparison to SARS-CoV-2/Wuhan, the maximum deviation in the Cα backbone of NTD was observed in SARS-CoV-2-β (0.45 Å) whereas in the case of RBD the maximum deviation was observed in SARS-CoV-2-o (0.14 Å).

**Figure 2 polymers-14-03263-f002:**
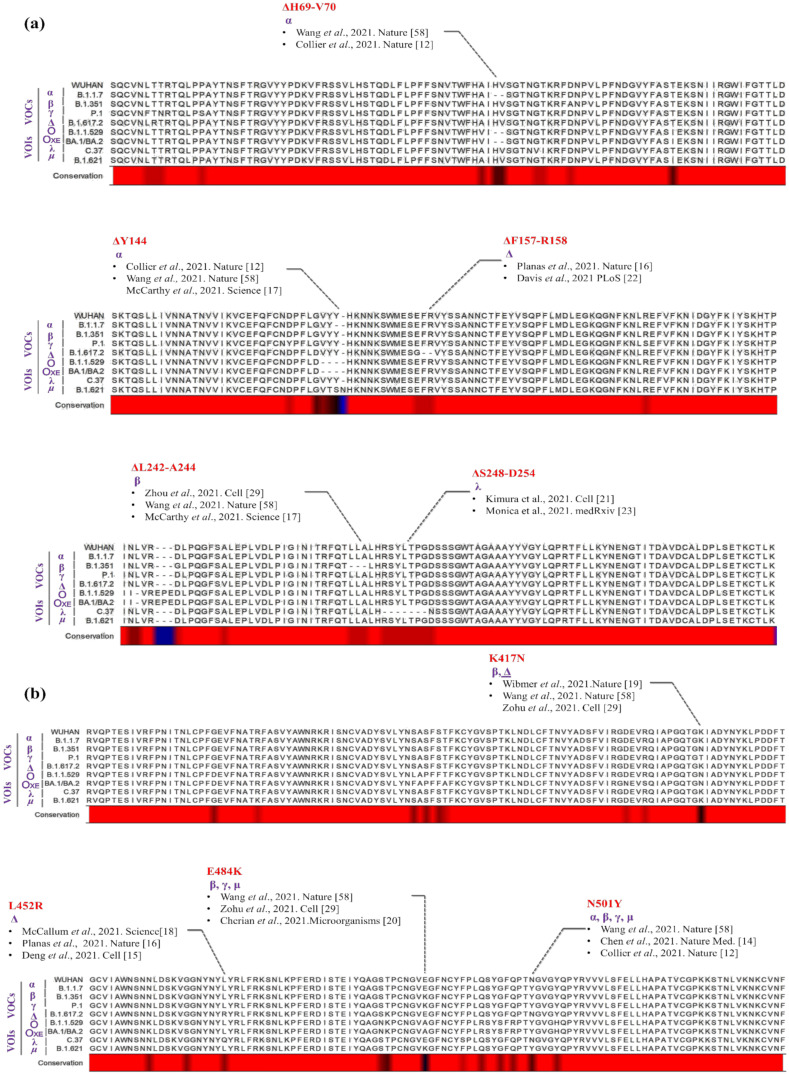
Multiple sequence alignment of NTDs and RBDs. Multiple sequence alignments of immunologically important regions (**a**) NTD (**b**) RBD of the SARS-CoV-2 VOCs and VOIs (as labeled) showing degree of sequence variations (black in red horizontal bar). Mutations known for their importance in the immune evasion are also highlighted with corresponding references [12,14,15,16,17,18,19,20,21,22,23,29,58].

### 3.3. Epitope Variations in NTDs and RBDs

Next, we explored the possibility that whether structural variations in the NTDs and RBDs of different SARS-CoV-2 variants influence the distribution and/or span of the epitopes in the domains. Structure-based epitope prediction showed that most of the epitopes in NTDs and RBDs are conserved between SARS-CoV-2 variants. However, the epitope, S71-N74, was found to vary between SARS-CoV-2/Wuhan and SARS-CoV-2-α and SARS-CoV-2-ο. Whereas, epitope, L237, and P239 of SARS-CoV-2/Wuhan were found marginally expanded in SARS-CoV-2-α, SARS-CoV-2-β, SARS-CoV-2-γ, and SARS-CoV-2-∆, and considerably expanded in SARS-CoV-2-ο. In total, this suggests that the antigenic potential of NTDs of different SARS-CoV-2 variants differs noticeably compared to SARS-CoV-2/Wuhan. Our hypothetically designed NTD (h-NTD) not only retains the structural conformation of NTD but importantly covers epitopes of all major variants of SARS-CoV-2 (Figure 3c; See Appendix A). Variations were also observed in the span of RBD epitopes amongst SARS-CoV-2 variants. Most noticeably, epitopes S443-N450 and G496-Y505 of SARS-CoV-2/Wuhan RBD were found marginally expanded in SARS-CoV-2-∆ and SARS-CoV-2-λ. Like h-NTD, our hypothetically designed RBD (h-RBD) also retains both structure and epitope diversity of the RBDs of major SARS-CoV-2 variants (Figure 3d; See Appendix A). Studies have shown reduced neutralization of many SARS-CoV-2 VOCs including omicron by post-vaccination sera [12,13,14,15,16,17,18,19,20,21,22,23,24,58,59]. Moreover, it has been shown previously that many of the critical mutations work in synergy in relation to the reduced neutralization of the SARS-CoV-2 variants by post-vaccination sera [12,13]. Considering the epitope map, assembling both h-NTD and h-RBD in a single construct could lead to the development of a vaccine that could provide protection against multiple immune escape variants of SARS-CoV-2. To further explore this possibility, we have modeled protein encoded by potential construct (DOW-21) by assembling both h-NTD and h-RBD, intervened by 10 mer of gly-ala repeat (Figure 4a). 

### 3.4. Structure and Epitope Prediction of mRNA Vaccine Constructs Encoded Proteins

In the SARS-CoV-2 spike protein, NTD and RBD are tandemly placed and known to adopt independent folding [57,60]. However, in the protein encoded by the construct, these domains are not in their natural combinations with respect to the full-length spike protein, therefore, could potentially adopt different structural conformation. This may further influence the distribution and span of the epitopes on the encoded protein. Moreover, the gly-ala linker could also become antigenic in three-dimensional conformations, especially in the presence of flanking domains (Figure 4a). To explore these possibilities, the full-length model of the protein encoded by the construct (DOW-21) was developed. Ramachandran plot and ProSA analyses showed that the modeled structure of DOW-21 was within an acceptable limit, indicating the structural plausibility of the protein (Figure 4b,c). Superimposition of the full-length protein over the NTD and RBD of SARS-CoV-2/Wuhan showed minimal Cα backbone deviation. This indicates that NTD and RBD in construct retained their structural conformation as observed in their individual states (Figure 4d,e). Consistently, the location and span of the epitopes were also found to be maintained by and large as observed in the h-NTD and h-RBD, covering all major epitopes found in the NTDs and RBDs of immune escape variants of SARS-CoV-2. Moreover, structure-based epitope prediction did not show any epitope and/or antigenic properties of the gly-ala unit within the protein translated from the DOW-21 construct (Figure 4f; See Appendix A).

### 3.5. Immune Simulation

The protein encoded by DOW-21 was predicted to elicit a substantial immune response in terms of IgG levels (~12,000 AU) and different cytokines at 15 days in C-ImmSim prediction. Amongst the cytokines, IFN-γ showed the peak level (~400,000 ng/mL) at 15 days of injection (Figure 5).

### 3.6. Multivariant COVID-19 mRNA Vaccine Construct

Since the DOW-21 protein is aimed to be encoded by mRNA, smooth translation of mRNA requires regulatory sequences. Therefore, in the mRNA construct of DOW-21, at the 5′ end, signal sequence and 5′-UTR sequence were added, whereas 3′end of the coding region was flanked by 3′-UTR and polyA tail as described in the methodology. The full-length sequence was then optimized for the GC (56%) content and verified again by in silico translation. The schematic representation of DOW-21 is shown in Figure 6.

### 3.7. MicroRNA-mRNA Vaccine Constructs Interactions

Screening of full-length mRNA sequence of construct, DOW-21, for miRNA binding site showed that DOW-21 has 24 potential miRNA binding sites below the threshold score of 90. Additionally, the highest score predicted by miRDB of potential miRNA (hsa-miR-6868-3p) that could bind with DOW-21 is 82 (Appendix A). The sequence file for DOW-21 is also made available (See Appendix A).

### 3.8. Structural Analysis of mRNA Vaccine Constructs

RNA fold analysis of optimized DOW-21 showed acceptable minimum free energy, centroid, and partition function across the length of the construct (Figure 6).

## 4. Discussion

The emergence of VOCs and VOIs of SARS-CoV-2, with many, have been shown as immune escape variants, potentially challenging the efficacy of currently available immunotherapeutic measures including vaccines against the virus [12,13,14,15,16,17,18,19,20,21,22,23,24,29,58]. Therefore, under the convention, multiple vaccines and/or multiepitope-based vaccines are needed to broaden the spectrum of protection against circulating SARS-CoV-2 variants. However, multiple vaccinations could in turn result in enhanced hypersensitivities and may suffer from non-compliance and cost constraints in relation to the logistics and vaccine production. Moreover, it would be challenging for the researchers and/or pharmaceuticals to abruptly respond to the demand for vaccines against the newly emerging SARS-CoV-2 variants, especially in developing countries. This issue could be addressed by designing a single multivariant vaccine that elicits the immune response against a wide range of SARS-CoV-2 variants. Given the flexibility and relative safety attributed to the mRNA vaccines, they provide the best platform in this regard [61,62].

Previously, it has been shown that two regions of the SARS-CoV-2 spike protein, NTD and RBD are the main target of the neutralizing antibodies elicited due to the natural infection of the virus and/or by vaccination [32,33]. It has been observed in different variants of SARS-CoV-2 especially in VOCs and VOIs that mutation in both NTD and RBD results in the development of escape variants of the virus [12,13,14,15,16,17,18,19,20,21,22,23,24,29,58]. In different SARS-CoV-2 variants, the new mutations are accumulated from the antecedent and convergently evolving strains of the virus (Figure 2). Resultantly, the epitope landscape of NTDs and RBDs in different SARS-CoV-2 variants has been changed (Figure 3). This connotes the need for alternate constructs of mRNA vaccine to provide broad coverage against immune escape variants of SARS-CoV-2. 

Previously, several studies, conducted in murine and non-human primates, have shown that alone RBD can elicit a significant immune response against SARS-CoV-2 [34,35]. Secondly, synergism in the mutations, particularly N501Y, E484K, and L452R, has been shown necessary in relation to immune evasion of SARS-CoV-2 variants [12,13]. Additionally, NTD was also found significant in neutralizing viral load [37,38]. This encourages us to develop an mRNA vaccine construct (DOW-21) containing hypothetically designed NTD (h-NTD) and RBD (h-RBD) (Figure 6). Previously, four SARS-CoV-2 variants, SARS-CoV-2-α, β, γ, ∆ have shown to exhibit noticeably reduced neutralization against the post-vaccination sera [12,13,14,15,16,17,18,19,20,21,22,23,29,58]. Recent reports for newly designated VOC, SARS-CoV-2-ο are also not very encouraging in this regard [24,31,59]. The designed h-NTD and h-RBD provide epitope coverage against all the variants of SARS-CoV-2 known for immune evasion, in form of a full-length construct (Figure 3 and Figure 4). To ensure the independent folding of NTD and RBD within the construct, a 10-mer-gly-ala repeat of EBNA1 protein is placed as a linker peptide. Linker peptides are the short amino acid sequences that are naturally present in several proteins to maintain the structural and functional independency of the domains. Previously, residues such as Pro, Arg, Phe, Thr, Glu, and Gln are often suggested as rigid spacers, however, glycine-rich regions are also observed as natural linkers [63]. In EBNA1, gly-ala repeats abridge GR1 and GR2 domains, responsible for the binding with EBP2 [48]. We have preferred gly-ala repeats over gly-pro because of several reasons. First, the natural tendency of independent folding of NTD and RBD does not require a rigid spacer [59,60]. Second, there is no report of immunogenicity of the gly-ala linker in EBV-infected individuals and our structure-based prediction also did not reveal any immunogenicity in the protein model of the full-length construct (Figure 4). Third, the gly-ala repeat inhibits proteasomal degradation of EBNA1 in a length-dependent manner [64]. Furthermore, it is interesting to note that mRNA vaccine translation is mostly desired to happen in APCs and/or B-cells. Gly-ala repeats have been shown to be critical in maintaining a relatively long half-life (>20 h) of the EBNA1 protein in B cells [65]. Finally, glycine-rich linkers have been shown not to influence the peptide recognition ability of MHC-II complexes [66]. 

Effective mRNA vaccine constructs also require regulatory sequences such as UTRs, Kozak sequence, signal sequence, and polyA tail to be adhered appropriately around encoding the antigen. Traditionally, globin UTRs are commonly used for in vitro transcription of mRNA and have been shown to induce the expression of the insert by 1000-fold [67]. A recent study has shown a de novo designed 5′-UTR along with conserved mammalian Kozak sequence and murine Rps27a 3′-UTR compounded with Sindbis virus RNA binding protein region, R3U, further improve the gene expression by a further 5- to 10-fold compared to globin UTR. This 5′-UTR sequence has also shown to have minimal secondary structure and reduced alternate translation initiation sites. Moreover, the same combination of 5′-UTR and 3′-UTR induces the expression of RBD of SARS-CoV-2 spike protein three-fold compared to α-polypeptide of cytochrome B [43]. Considering this, we have used the same combination of 5′-UTR and 3′-UTR in our constructs. The signal sequence is another important component of the mRNA vaccine as it mediates the intracellular transportation of the molecule. A comparison of signal sequences derived from human Igκ and Ebola virus glycoprotein in mRNA constructs has shown 12-fold increments with the former in the titer of neutralizing antibodies [44]. Thereby, in our construct, we have used a signal sequence from human Igκ for intracellular transportation of the construct. MicroRNAs post-transcriptionally modulate gene expression in eukaryotes [68]. Therefore, it is important that the expression of mRNA constructs must not be affected by the miRNA. In the miRNA binding site screening of our construct (DOW-21), very few miRNA binding sites were present in terms of scoring and number of sites (Appendix A). Vaccines including mRNA vaccines essentially require not to elicit any autoimmune response. It has been predicted that 23 of the peptides derived from different proteins of SARS-CoV-2 have sequence homology with different human genes. Out of these twenty-three peptides, two are present within the spike protein of the virus [69]. Sequence alignment of the protein encoded by our construct (DOW-21) with these 23 peptides and especially with two of the spike protein peptides showed no sequence homology, therefore may not induce any unwanted autoimmune response. This also further rationalizes the use of a domain-specific construct for mRNA vaccine. 

The continuous emergence of SARS-CoV-2 variants, especially those with the potential for immune escape and/or evasion demands a vaccine platform that has enough ingrained flexibility to respond to the challenge. In this regard mRNA vaccine holds a considerable advantage over conventional vaccines in that simple alteration of the sequence can express new and/or emerging antigens that could then be channeled into an already established downstream processing system [61]. Moreover, mRNA vaccine candidates have been clinically tested for their safety against HIV-1, Zika virus, rabies, influenza virus, and cancers with encouraging results [7,8,9,10,70]. However, like all computationally designed molecules for clinical applications, the proposed construct essentially warrants experimental validation. Nevertheless, the immunoinformatic approaches are in continuous use to facilitate the designing and development of both conventional and mRNA-based vaccines [71,72,73]. In addition, there are certain hurdles and potential side effects associated with mRNA vaccines. First and perhaps foremost is the stability of the mRNA, particularly for the countries that lack proper facilities for cold chain transportation and storage. To improve the stability of mRNA vaccines, several approaches such as incrementing the proportion of GC or incorporation of histone stem-loop have been employed [74,75]. The GC ratio of DOW-21 is also within the acceptable limit. Contaminant in the mRNA vaccines, particularly dsDNA/RNA, is a pathogen-associated molecular pattern (PAMP) molecule, which could trigger an unwanted innate immune response, particularly of type I interferon [76]. This issue could be mitigated by optimizing the purification strategies [77]. Moreover, instead of using conventional uracil, N1-methyl-pseudouridine also addresses this shortcoming [78]. 

To date, two mRNA-based vaccines for SARS-CoV-2 have been used for mass vaccination. The delivery vehicle for both these vaccines is primarily lipid nanoparticles with varying chemical nature and proportion of components [79]. These vehicles have shown effective delivery of the nucleic acid molecule without any major adverse effects. Therefore, for the proposed construct, DOW-21, the same or similar lipid-based nanoparticle vehicle could be used for encapsulation and delivery of the vaccine candidate. The mRNA construct may then elicit the immune response in a similar fashion as observed for other SARS-CoV-2 mRNA vaccines, where mRNA molecules are taken up by antigen-presenting cells to express the antigen molecule on the surface. This in turn results in the surge of cytokine-mediated innate immune response and later translated into the T and B cell-mediated humoral immune response [80]. Nevertheless, optimization of the delivery system and elucidation of the mode of action of DOW-21 necessarily require additional empirical investigations. 

## 5. Conclusions

Despite the presence of preventive vaccines against SARS-CoV-2, the threat of COVID-19 is not subsiding due to the continuous emergence of immune escape variants of the virus. The recent surge of omicron variant infection is yet another example in this regard. The mRNA vaccine constructs presented in this study are novel both in terms of approach and design. Instead of using a whole length of spike protein or its RBD domain of SARS-CoV-2, we have used both immunogenic regions (NTD and RBD domains) of the viral spike protein. The designed mRNA construct comprises NTD and RBD in a single molecule where each of the domains contains known immune escape mutations of the viral spike protein. The protein encoded by the construct retains the structural conformation of the domains but exhibits diversity in epitopes that cover antigen variations found in different variants of the virus. That is a construct configuration that is expected to provide protection against multiple immune escape variants of SARS-CoV-2.

## Figures and Tables

**Figure 1 polymers-14-03263-f001:**
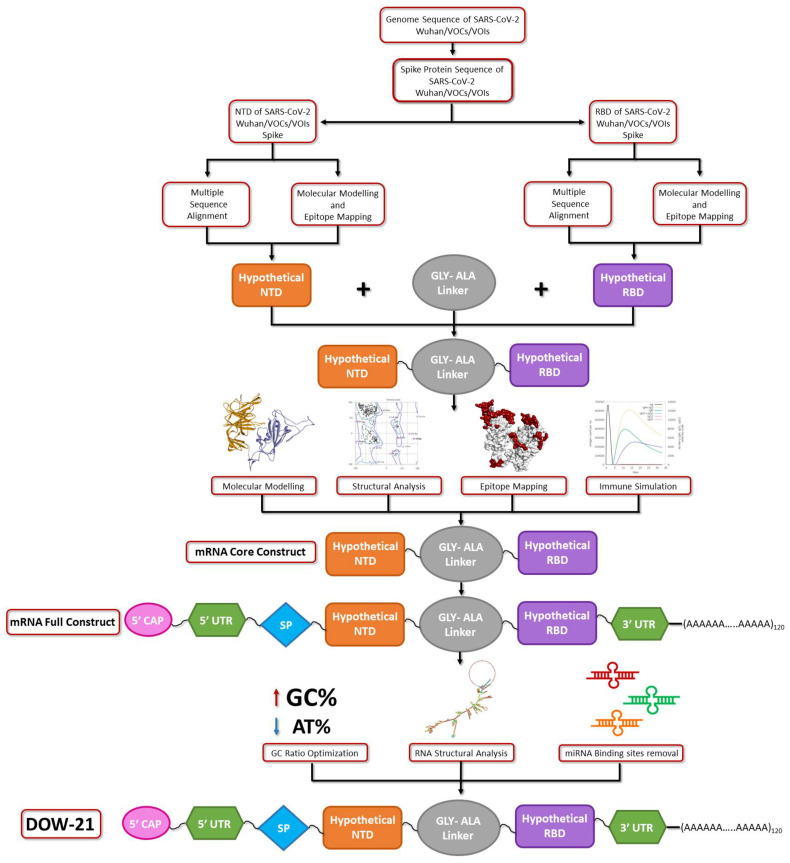
Flow chart of methodology adopted in the present investigation. Key: NTD: N-terminal domain, RBD: receptor binding domain, SP: signal peptide, UTR: untranslated region.

**Figure 3 polymers-14-03263-f003:**
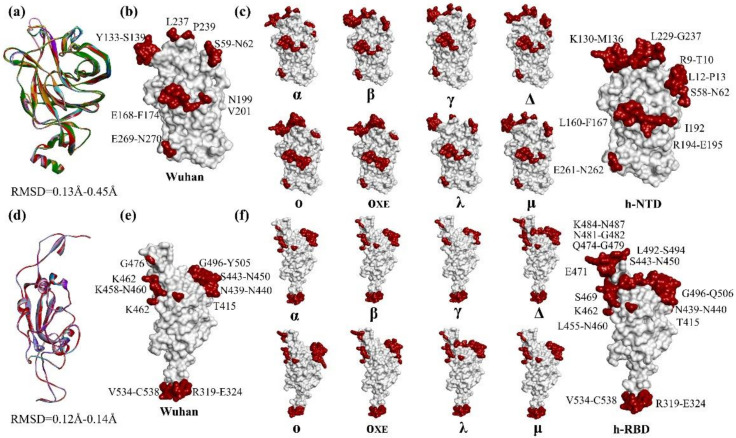
Structure and epitopes of NTDs and RBDs. (**a**) Ribbon diagram of superimposed NTD of different SARS-CoV-2 variants including h-NTD showing structural variations. Distribution of epitope (maroon) in the surface topology of NTD of (**b**) SARS-CoV-2/Wuhan and (**c**) different SARS-CoV-2 VOCs, VOI, and h-NTD as labeled. (**d**) Ribbon diagram of superimposed RBD of different SARS-CoV-2 variants including h-RBD representing structural variations. Distribution of epitope (maroon) in the surface topology of RBD of (**e**) SARS-CoV-2/Wuhan and (**f**) different SARS-CoV-2 VOCs, VOI, and h-RBD as labeled.

**Figure 4 polymers-14-03263-f004:**
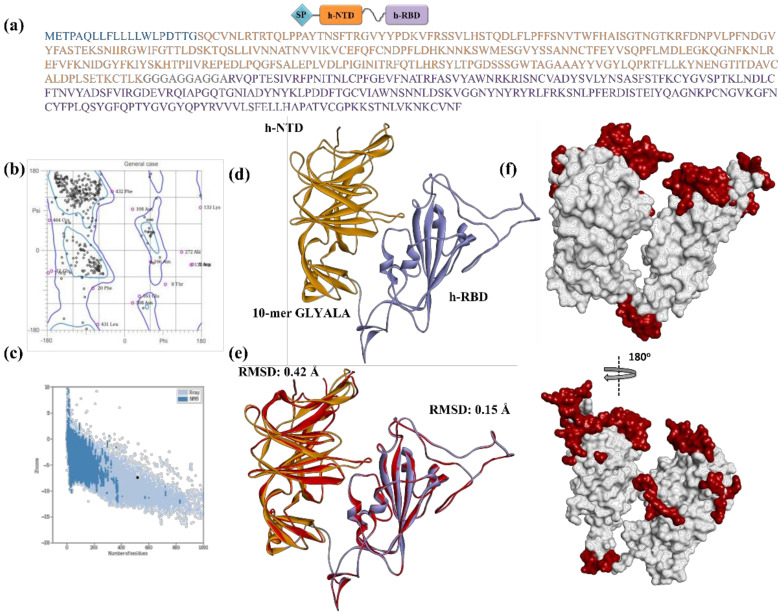
Sequence and structure of designed mRNA construct encoded protein (DOW-21). (**a**) Schematic representation and amino acid sequence of designed mRNA encoded protein (DOW-21) showing distribution of signal sequence (cyan), h-NTD (orange), and h-RBD (purple), intervened by gly-ala linker. (**b**) Ramachandran plot showing distribution of dihedral angles (black dots) of DOW-21 protein model in the favored (blue contour) and allowed (purple contour) regions. (**c**) Z-Score of DOW-21 model (black filled circle) in the Z-score graph of empirically resolved protein structures deduced by NMR (dark blue) and X-ray crystallography (light blue). (**d**) Ribbon diagram of DOW-21 model showing h-NTD, gly ala linker, and h-RBD in corresponding colors. (**e**) Ribbon diagram of DOW-21 superimposed with SARS-CoV-2/Wuhan NTD and RBD with RMSD values mentioned. (**f**) Distribution of epitope (maroon) in the surface topology of DOW-21 with 180^o^ rotation showing retention of the antigenic regions as found in the NTD and RBD of multiple SARS-CoV-2 variants.

**Figure 5 polymers-14-03263-f005:**
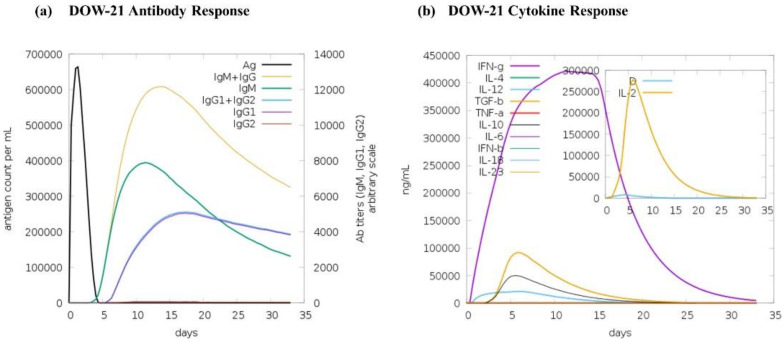
Immune simulation of DOW-21. Predicted immune response for (**a**) antibodies and (**b**) cytokines along with danger signal (inset) of DOW-21.

**Figure 6 polymers-14-03263-f006:**
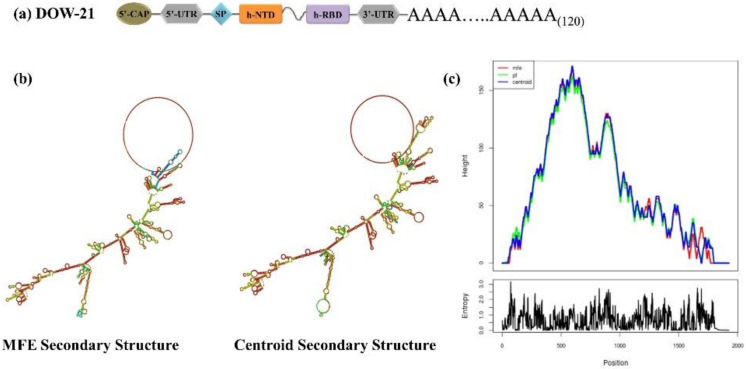
Predicted secondary RNA structure of DOW-21 construct. (**a**) Schematic representation of full-length mRNA construct; (**b**) secondary structures and (**c**) mountain plot of DOW-21. Green, red, and blue lines in the mountain plot represent partition function, minimum free energy, and centroid, respectively.

## Data Availability

The data presented in this study are available in Appendix A.

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
