# Peer review of "Design and Immunoinformatic Assessment of Candidate Multivariant mRNA Vaccine Construct against Immune Escape Variants of SARS-CoV-2"

_polymers, 2022, doi:10.3390/polym14163263_

Round 1

Reviewer 1 Report

 This manuscript “Design and Immunoinformatic Assessment of Candidate Multivariant mRNA Vaccine Construct Against Immune Escape Variants of SARS-CoV-2” prepared by Hussain et al. presented the potential to design a distinct construct as a candidate for the SARS-CoV-2 vaccine. The analysis of spike protein and potential human response is greatly appreciated, and revisions clarifying the importance of the findings will be needed.

In method 2.2: what setting did the authors use for CLUSTALW alignment, please clarify.

Author Response

Point 1. This manuscript “Design and Immunoinformatic Assessment of Candidate Multivariant mRNA Vaccine Construct Against Immune Escape Variants of SARS-CoV-2” prepared by Hussain et al. presented the potential to design a distinct construct as a candidate for the SARS-CoV-2 vaccine. The analysis of spike protein and potential human response is greatly appreciated, and revisions clarifying the importance of the findings will be needed.

In method 2.2: what setting did the authors use for CLUSTALW alignment, please clarify.

Response: We are grateful to the reviewer for an important observation that may be needed to replicate the alignment of NTDs and RBDs of SARS-CoV-2 immune escape variants. The alignments were carried out using Clustal W under default parameters and the point is now been added in the revised version of the manuscript. 

Reviewer 2 Report

Authors should explain and include the following comments in the revised manuscript

1.      Prediction and assessment of NTD and RBD sequences

2.      Prediction of antigenicity, allergenicity, toxicity, and physicochemical properties of the vaccine construct

3.      For immune simulations, how many injections, and how many vaccines construct per units

4.      Evaluation of the simulated immune response against the vaccine

5.      Proposed mechanism of delivery and action of the mRNA vaccine against SARS-CoV-2

Author Response

Point 1. Prediction and assessment of NTD and RBD sequences

Response: We are thankful for the reviewer note. Methods employed for the prediction and assessment of the NTDs and RBDs of different SARS-CoV-2 variants are described in 2.1 and 2.2. Whereas the results of the analysis are presented in 3.1 to 3.3. 

Point 2. Prediction of antigenicity, allergenicity, toxicity, and physicochemical properties of the vaccine construct

Response: We are grateful to the reviewer. Methods used in order to predict the antigenicity and physicochemical properties of the vaccine construct are described from section 2.7 to 2.12. Whereas the results of the analysis are presented in 3.4 to 3.8.  Section 3.4 has been modified to better represent the results. 

Point 3. For immune simulations, how many injections, and how many vaccines construct per units

Response: This indeed is a very crucial information require to replicate and empirically assess the prediction made in our study and we are grateful to the reviewer for highlighting this important point. Immune simulation prediction of DOW-21 construct is based on single injection comprising 1000 mRNA molecules per unit. The information is also mentioned in our revised manuscript 

Point 4. Evaluation of the simulated immune response against the vaccine

Response: We are highly grateful to the reviewer for highlighting this important point. The simulated immune response predicted to elicit antibody titre of 12,000 AU after 15 days of injection. At the same time IFNγ also predicted to surge upto 400000 ng/ml. This demonstrates a potential robust immune response against the proposed mRNA construct. However, this prediction needs further empirical validation, that has been discussed as a potential limitation in the manuscript.

Point 5. Proposed mechanism of delivery and action of the mRNA vaccine against SARS-CoV-2

Response: We are thankful to the reviewer for suggesting the inclusion of proposed mechanism of delivery and action of mRNA vaccine in the manuscript. This will certainly strengthen the draft quality. In our revised manuscript we have added a paragraph at the end of discussion in this regard. 

Reviewer 3 Report

The article is exciting work and is the future for new vaccine for SARS-COV-2 but the work needs extra work to prove the theory of this work.

The article titled Design and Immunoinformatic Assessment of Candidate Mul-tivariant mRNA Vaccine Construct Against Immune Escape Variants of SARS-CoV-2 after consideration of minor comments.

.

1)      Abstract, Some abbreviations should be explain firstly appear.

2)      Abstract ,  it is not better to use we , I , US

3)      Authors should add results to abstract

4)      Introduction, the rational  for this study should be  improved.

5)      Conclusion should be improved  ?

6)      It is better to merge results and discussion.

Author Response

Point 1. Abstract, Some abbreviations should be explain firstly appear.

Response: We thank reviewer for this observation. We have modified the abstract in the revised manuscript as per the reviewer suggestion. 

Point 2.  Abstract ,  it is not better to use we , I , US

Response: We thank reviewer for this note.  The abstract has been modified as per the reviewer suggestion. 

Point 3. Authors should add results to abstract

Response: We are thankful to the reviewer for this valuable suggestion to strengthen the draft of the abstract. Due to the limitation (as per journal format) of 200 words, it is rather difficult to summarised all the results. However, few more key results (quantitative) have been incorporated in the abstract of the revised manuscript. 

Point 4. Introduction, the rational  for this study should be  improved.

Response: We thank reviewer for this note. In the revised manuscript some modifications have been made in  wording the rationale to improve the clarity. 

Point 5 Conclusion should be improved  ?

Response: We are thankful to the reviewer for this suggestion. In the revised manuscript some modifications have been made in the conclusion. 

Point 6. It is better to merge results and discussion

Response: We agree with the esteemed reviewer suggestion that results and discussion could be merged in the manuscript. However, respectfully, we would prefer to draft them separately, which is also in line with the journal's recommended format.  

Round 2

Reviewer 2 Report

Accept in present form

Author Response

Point 1: Accept in present form 

Response: We are grateful to the esteemed reviewer.